

# Sequence characteristics and phylogenetic analysis of H9N2 subtype avian influenza A viruses detected from poultry and the environment in China, 2018

Xiaoyi Gao[1,2], Naidi Wang[1], Yuhong Chen[3], Xiaoxue Gu[1], Yuanhui Huang[3], Yang Liu[1], Fei Jiang[1], Jie Bai[1], Lu Qi[1], Shengpeng Xin[1], Yuxiang Shi[2], Chuanbin Wang[1] and Yuliang Liu[1]

[1] National Veterinary Diagnostic Center, China Animal Disease Control Center, Beijing, P.R.China
[2] College of Life Sciences and Food Engineering, Hebei University of Engineering, Handan, Hebei, P.R.China
[3] College of Animal Science and Technology, GuangXi University, Nanning, Guangxi, P.R.China

Corresponding authors
Chuanbin Wang,
nvdcwang@sina.com
Yuliang Liu, ylliu0905@163.com

## ABSTRACT

H9N2 subtype avian influenza A virus (AIV) is a causative agent that poses serious threats to both the poultry industry and global public health. In this study, we performed active surveillance to identify H9N2 AIVs from poultry (chicken, duck, and goose) and the environment of different regions in China, and we phylogenetically characterized the sequences. AIV subtype-specific reverse transcription polymerase chain reaction (RT-PCR) showed that 5.43% (83/1529) samples were AIV positive, and 87.02% (67/77) of which were H9N2 AIVs. Phylogenetic analysis revealed that all H9N2 field viruses belonged to the Y280-like lineage, exhibiting 93.9–100% and 94.6–100% of homology in the hemagglutinin (HA) gene and 94.4–100% and 96.3–100% in the neuraminidase (NA) gene, at the nucleotide (nt) and amino acid (aa) levels, respectively. All field viruses shared relatively lower identities with vaccine strains, ranging from 89.4% to 97.7%. The aa sequence at the cleavage site (aa 333–340) in HA of all the isolated H9N2 AIVs was PSRSSRG/L, which is a characteristic of low pathogenic avian influenza virus (LPAIV). Notably, all the H9N2 field viruses harbored eight glycosylation sites, whereas a glycosylation site 218 NRT was missing and a new site 313 NCS was inserted. All field viruses had NGLMR as their receptor binding sites (RBS) at aa position 224–229, showing high conservation with many recently-isolated H9N2 strains. All H9N2 field isolates at position 226 had the aa Leucine (L), indicating their ability to bind to sialic acid (SA) α, a 2–6 receptor of mammals that poses the potential risk of transmission to humans. Our results suggest that H9N2 AIVs circulating in poultry populations that have genetic variation and the potential of infecting mammalian species are of great significance when monitoring H9N2 AIVs in China.

Subjects Molecular Biology, Veterinary Medicine, Virology, Infectious Diseases
Keywords H9N2 subtype, Avian influenza virus, Phylogenetic analysis, Epidemiology

## INTRODUCTION

Avian influenza A virus (AIV), belonging to the family *Orthomyxoviridae*, carries a segmented, single-stranded, negative sense RNA genome with enveloped virions. The

AIV genome contains eight segments with the ability to code various proteins: polymerase proteins (PB1, PB2, PA), nucleocapsid protein (NP), hemagglutinin (HA), neuraminidase (NA), matrix protein (M), and nonstructural protein (NS) (*Jagger et al., 2012*; *Selman et al., 2012*; *Wise et al., 2012*). Based on their virulence in chickens, AIVs can be categorized into low pathogenic avian influenza viruses (LPAIVs) and highly pathogenic influenza viruses (HPAIVs).

Chinese H9N2 LPAIV was first isolated from diseased chickens in Beijing in 1994 (*Chen & Zhang, 1997*; *Huang et al., 2010*). Subsequently, H9N2 subtype AIV spread rapidly across China, posing a high risk to domestic poultry, special economic animals, and, occasionally, humans (*Butt et al., 2005*; *Wang et al., 2018*; *Xu et al., 2007*; *Peiris, Poon & Guan, 2012*; *Zhang et al., 2019*). H9N2 subtype AIV can be transmitted and disseminated via the respiratory tract and contaminated wild bird feces, further complicating the control of this disease. In general, poultry infected with H9N2 virus only exhibit mild respiratory symptoms, but the wide spread of H9N2 subtype AIV could cause significant economic losses due to declined egg production, lower growth performance, increased risk of immunosuppression, and high mortality caused by coinfection with other respiratory pathogens (*Liu et al., 2003*). It has been documented that H9N2 AIV donates its inner genes to the lethal H5N1 influenza virus, as well as the H7N9 and H10N8 influenza virus, which could potentially lead to a human pandemic (*Babakir-Mina et al., 2014*; *Li, Shi & Chen, 2014*; *Cui et al., 2020*; *Guan et al., 1999*; *Pu et al., 2015*).

Currently, vaccination is the most effective strategy for H9N2 AIV prevention and control implemented in China and many other countries around the world. However, genetic recombination and antigenic variation are the most crucial explanations for mismatch between vaccine strains and dominant circulating strains, which at least partly explains the reduced efficacy of commercial vaccines against H9N2 AIV. Because of this, it is essential to continuously monitor the genetic evolution of H9N2 viruses. Previous studies found that the H9N2 virus preferred to bind human-type sialic acid (SA) receptors, and that the number of humans infected with H9N2 AIV has increased in China in recent years (*Li et al., 2017*). The universal method classified H9N2 viruses into two major lineages, the North American lineage and the Eurasian lineage. The Eurasian lineage consists of four lineages: KR323-like (*e.g.,* A/Chicken/Korea/38349-p96323/96), Y439-like (*e.g.,* A/Duck/Hong Kong/Y439/1997), Y280-like (*e.g.,* A/Duck/Hongkong/Y439/97), and G1-like (*e.g.,* A/Quail /Hong Kong/G1/97) (*Guan et al., 1999*). The Y280-like lineage was made up of five clades: A/chicken/Beijing/1/94-like (BJ/94-like), A/Chicken/Shandong/1998 (SD98-like), A/Chicken/HongKong/G9/1997 (G9-like), A/Chicken/Shanghai/F/1998 (SH98-like), and Y280-like. The BJ94-like lineage was substituted with the SH98-like lineage starting in 2004 (*Huang et al,. 2010*). The other method was established based on the evolutionary distance between the HA sequences of wild type isolates and classical strains. It was recently reported that the lineages h9.4.2.5 and h9.4.2.6 were dominant in China (*Chen et al., 2013*; *Jiang et al., 2017*; *Shen et al., 2015*).

The AIV HA protein is a critical antigen associated with hemagglutinating activity and the adsorption and penetration of viral infections (*Post et al., 2013*; *Xia et al., 2017*). The NA protein mainly promotes the release of virus particles and prevents the aggregation

of progeny virus particles by hydrolyzing sialic acid residues on the surface of viruses and cells. In this study, we conducted an active surveillance of AIV. Using RT-PCR, sequencing, virus isolation, and genetic analysis software, we analyzed the HA and NA of H9N2 subtype AIVs in different poultry species and environmental samples from different provinces in the eastern regions of China, aiming to explore their genetic relationship. We detected genetic variations in the H9N2 viruses, especially in the HA gene, from healthy poultry. Therefore, we suggest that the continuous surveillance of H9N2 AIVs in poultry farms be enhanced.

## MATERIALS AND METHODS

### Samples and treatment

Field sample collection was approved by the Chinese Animal Disease Control Center (CADC (surveillance) (2018) No. 53). A total of 1,529 oropharyngeal and cloacal swab samples were collected from healthy chickens, ducks, geese, pigeons, and the environment from the live bird markets (LBM) in Shandong, Zhejiang, Hubei, and Jiangxi provinces, China, in 2018. Of the 1,529 samples, 20 were collected from the LBM environment. The swabs were placed into PBS containing 1‰ penicillin-streptomycin, stored in a cold environment with ice bags, transported to the laboratory within 24 h, and immediately frozen at $-80\,°C$ before detection. The detailed information about the collected samples is shown in Table 1. All swabs were centrifuged at 3,000 rpm for 10 min at 4 °C, and the supernatant of each sample was harvested and used for RNA extraction and subsequent RT-PCR. The AIV in the samples that were detected to be positive for the H9N2 subtype by RT-PCR and sequencing were inoculated into 9-day-old specific-pathogen-free (SPF) embryonated chicken eggs (Beijing Merial Vital Laboratory Animal Technology Co., Ltd., Beijing, China) at a volume of 0.2 mL/egg. The inoculated chicken embryos were incubated at 37 °C at a humidity of 50–60% and were monitored every day. The embryonated chicken eggs died between 24 and 144 h post-inoculation (hpi), were timely collected, and were chilled for 24 h. Subsequently, the allantoic fluid was harvested from the allantoic cavities of all embryonated eggs and purified by centrifugation at 4 °C and 3,000× rpm for 10 min. The allantoic fluid was then used for hemagglutination and hemagglutination inhibition (HI) tests.

### RNA extraction, RT-PCR, and sequencing

Viral RNA was extracted using the QIAamp Viral RNA Mini Kit (Qiagen, Hilden, Germany) according to the manufacturer's protocol. Primers were synthesized by Genewiz Co. Ltd., Suzhou, China. First, the partial NP gene fragment, with an expected size of 330 bp, was amplified according to the Agricultural Standard of China (standard ID: NY/T 772-2013). The sequence of the forward primers used was 5′-CAGRTACTGGGCHATAAGRAC-3′and the reverse primer sequence was 5′-GCATTGTCTCCGAAGAAATAAG-3′. A reaction system with a volume of 20 μL was prepared for one-step RT-PCR, consisting of 10 μL of 2× buffer, 0.8 μL of $Taq^{TM}$ polymerase, 0.8 μL of forward and reverse primer mix (10 pmol/μL each), 0.2 μL of AMV reverse transcriptase (TaKaRa, Japan), 0.2 μL of RNasin ribonuclease inhibitor (Promega, Madison, USA), 2 μL of RNA template, and 6 μL of
**Table 1 Geographic distribution and source species of H9N2 avian influenza viruses.**

| No. | Name of virus isolate | Species |
|---|---|---|
| 1 | A/Chicken/Hubei/76/2018 | Chicken |
| 2 | A/Chicken/Jiangxi/C22/2017 | Chicken |
| 2 | A/Chicken/Hubei/81/2018 | Chicken |
| 3 | A/Chicken/Hubei/61/2018 | Chicken |
| 4 | A/Chicken/Hubei/127/2018 | Chicken |
| 5 | A/Chicken/Shandong/C84/2018 | Chicken |
| 6 | A/Duck/Jiangxi/D10/2018 | Duck |
| 7 | A/Chicken/Shandong/C131/2018 | Chicken |
| 8 | A/Chicken/Hubei/226/2018 | Chicken |
| 9 | A/Chicken/Hubei/142/2018 | Chicken |
| 10 | A/Chicken/Hubei/126/2018 | Chicken |
| 11 | A/Chicken/Hubei/147/2018 | Chicken |
| 12 | A/Chicken/Hubei/149/2018 | Chicken |
| 13 | A/Chicken/Hubei/251/2018 | Chicken |
| 14 | A/Chicken/Shandong/C54/2018 | Chicken |
| 15 | A/Chicken/Hubei/95/2018 | Chicken |
| 16 | A/Chicken/Hubei/261/2018 | Chicken |
| 17 | A/Chicken/Hubei/80/2018 | Chicken |
| 18 | A/Chicken/Hubei/160/2018 | Chicken |
| 19 | A/Chicken/Shandong/C107/2018 | Chicken |
| 20 | A/Chicken/Shandong/C169/2018 | Chicken |
| 21 | A/Chicken/Jiangxi/C15/2018 | Chicken |
| 22 | A/Chicken/Shandong/C79/2018 | Chicken |
| 23 | A/Environment/Jiangxi/E15/2018 | Environment |
| 24 | A/Chicken/Jiangxi/C19/2018 | Chicken |
| 25 | A/Chicken/Shandong/C80/2018 | Chicken |
| 26 | A/Chicken/Jiangxi/C16/2018 | Chicken |
| 27 | A/Chicken/Hubei/146/2018 | Chicken |
| 28 | A/Chicken/Hubei/78/2018 | Chicken |
| 29 | A/Chicken/Jiangxi/C11/2018 | Chicken |
| 30 | A/Chicken/Hubei/131/2018 | Chicken |
| 31 | A/Chicken/Hubei/92/2018 | Chicken |
| 32 | A/Chicken/Jiangxi/C14/2018 | Chicken |
| 33 | A/Chicken/Hubei/63/2018 | Chicken |
| 34 | A/Chicken/Shandong/C126/2018 | Chicken |
| 35 | A/Chicken/Hubei/65/2018 | Chicken |
| 36 | A/Chicken/Hubei/107/2018 | Chicken |
| 37 | A/Duck/Jiangxi/D1/2018 | Duck |
| 38 | A/Chicken/Hubei/90/2018 | Chicken |
| 39 | A/Chicken/Hubei/169/2018 | Chicken |
| 40 | A/Chicken/Hubei/77/2018 | Chicken |
| 41 | A/Chicken/Jiangxi/C42/2018 | Chicken |

| No. | Name of virus isolate | Species |
|-----|----------------------|---------|
| 42 | A/Goose/Jiangxi/G1/2018 | Goose |
| 43 | A/Chicken/Hubei/152/2018 | Chicken |
| 44 | A/Chicken/Hubei/262/2018 | Chicken |
| 45 | A/Chicken/Shandong/C82/2018 | Chicken |
| 46 | A/Chicken/Hubei/251/2018 | Chicken |
| 47 | A/Duck/Jiangxi/D13/2018 | Duck |
| 48 | A/Chicken/Shandong/C131/2018 | Chicken |
| 49 | A/Chicken/Hubei/85/2018 | Chicken |

nuclease-free water (TaKaRa, Japan). All reagents were mixed gently and one step RT-PCR was conducted by incubating the mixture at 42 °C for 45 min for reverse-transcription (RT), followed by initial denaturation at 94 °C for 3 min, and 35 cycles of 94 °C for 30 s, 52 °C for 30 s, and 72 °C for 30 s, and final extension was 72 °C for 8 min. All AIV-positive samples were selected and further detected using one-step RT-PCR and primers Bm-HA-F (5′-TATTCGTCTCAGGGAGCAAAGCAGGG-3′) and Bm-HA-R (5′-ATATCGTCTCGTATTAGTAGAAACAAGGGTGTTTT-3′) to amplify the open reading frame (ORF) of the AIV HA gene fragment with an expected size of approximately 1.7 kb, as well as primers Ba-NA-1 (5′-TATTGGTCTCAGGGAGCAAAAGCAGGAGT-3′) and Ba-NA-R (5′-ATATGGTCTCGTATTAGTAGAAACAAGGAGTTTTTT-3′) to amplify the ORF of the NA gene fragment with an expected size of approximately 1.4 kb (*Hoffmann et al., 2001*). Briefly, a reaction system with a volume of 50 µL was prepared for RT-PCR, consisting of 25 µL of Premix *Taq*^TM (TaKaRa, Japan), 0.4 µL of AMV reverse transcriptase (TaKaRa, Japan), 0.6 µL of RNasin ribonuclease inhibitor (Promega, Madison, USA), 0.1 µL of reverse primer, 0.1 µL of forward primer, 18.2 µL of nuclease-free water (TaKaRa, Japan), and 4 µL of RNA. All reagents were mixed gently and incubated at 42 °C for 45 min for RT. The PCR program was initial denaturation at 95 °C for 3 min, followed by 35 cycles of 94 °C for 30 s, 56 °C for 30 s, and 68 °C for 2 min 30 s, and final extension was 72 °C for 10 min. Allantoic fluid harvested from embryonated SPF chicken eggs (purchased from Charles River, Beijing, China) and H9N2 subtype avian influenza virus strain A/Turkey/Wisconsin/1/1966 (purchased from ATCC) were used as negative and positive controls for the aforementioned RT-PCR, respectively. RT-PCR products were purified using a MiniBEST Agarose Gel DNA Extraction Kit (TaKaRa, Japan). The purified RT-PCR products were quantified using a Nanodrop 2000 (Thermo Fisher) and sent to Genewiz Co. Ltd. (Suzhou, China) for sequencing.

## Phylogenetic analysis

The nucleotide (nt) and amino acid (aa) homologies were assessed using the MegAlign Clustal W method in MegAlign. MEGA7.0 was used for phylogenetic tree construction using the neighbor-joining method with Kimura's two-parameter distance model and 1,000 bootstrap replicates (*Xia et al., 2017*). The trees included recently-isolated H9N2 AIV strains from China, major ancestral H9N2 AIV strains, and other reference H9N2 AIV strains. Reference sequences were obtained from the Global Initiative on Sharing All

Influenza Data (GISAID) and GenBank. BioEdit program version 7.0 was used for genomic analysis.

## Antigenicity analysis

The antigenicity of the selected field H9N2 isolates was detected using HA and HI tests.

# RESULTS

## RT-PCR

To conduct surveillance on H9N2 subtype AIVs, a total of 1,529 pharyngeal and cloacal swabs, as well as environmental LBM samples, were collected from Zhejiang, Hubei, Shandong, and Jiangxi provinces, China. RT-PCR detection of these samples determined that 83 isolates were positive for AIV, and 87.02% (67/77) of AIV-positive isolates were H9N2 subtypes, while Sanger sequencing determined that 6.49% (5/77), 5.19% (4/77), and 1.30% (1/77) samples were H7N9, H3N6, and H4N6 subtypes, respectively. It should be noted that no HA and NA gene fragments could be amplified in six AIV-positive samples for unknown reasons. No H9N2 AIVs were detected in the 381 samples from Zhejiang province, and the majority of H9N2 AIV positive samples (56.82% (24/44)) came from Hubei province. Our results showed that 25.00% and 18.18% of H9N2 AIV positive samples originated from Shandong and Jiangxi provinces, respectively. Notably, all H9N2 AIV-positive samples were isolated from chickens, except for two viruses found in ducks, one virus from a goose, and one from the environment.

## Phylogenetic analysis of HA

We performed sequence alignment and analysis of HA and NA of all isolated H9N2 AIVs. As shown in Fig. 1, all H9N2 viruses belonged to the Y280-like lineage whose representative strain was A/Duck/HongKong/Y280/97. Overall, the HA homologies at the nt and aa levels were 93.6–100% and 93.8–100%, respectively (Fig. S1A). Interestingly, all field H9N2 viruses shared relatively lower identities of 89.4–90.6%, 91.4–92.2%, 94.5–95.7%, 94.6–95.4%, 95.7–97.7%, and 95.7–97.7% for the vaccine strains A/Chicken/Guangdong/SS/94, A/Chicken/Shandong/YB06/2006, A/Chicken/Guangdong/Y4/2009, A/Chicken/Shandong/HL/2010, A/Chicken/Jiangsu/WJ57/2012, and A/Chicken/Jiangsu/SZ0207/2013, respectively (Fig. S1B). A phylogenetic tree was constructed based on the HA sequences, utilizing a top 30% cutoff. The results indicated that all isolated H9N2 viruses were divided into three distinct clusters (Fig. 1A). A total of 17 H9N2 viruses, ten of which originated from Shandong and seven from Hubei, were clustered into a separate branch with a lower homology rate compared to others. Interestingly, our results demonstrated further genetic distance between the isolated strains and the representative Y280 strain or the vaccine strains (Fig. 1). H9N2 viruses isolated from Hubei and Jiangxi were divided into two clusters and, compared to the isolated strains from recent years, a further genetic relationship was indicated between the field strains.

The cleavage site (aa 333–340) in the HA of all the H9N2 isolates was PSRSSR/GL, a typical characteristic of LPAIV (data not shown). A total of eight potential glycosylation sites were identified in the HA of all isolates. Interestingly, a 218NRT site was missing and

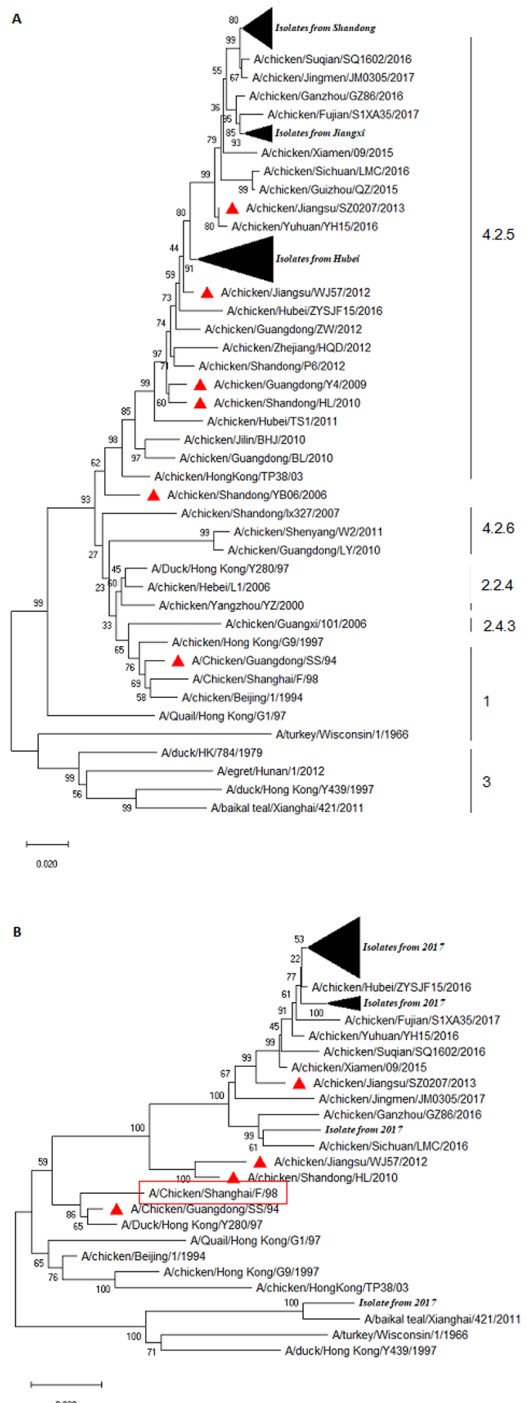

**Figure 1** **Phylogenetic relationship of the HA (A) and NA (B) genes of H9N2 AIVs detect in this study and reference strains.** The tree was constructed by neighbor-joining method using MEGA package version 7.0. Note: (A) the HA sequences of prevalent H9N2 isolates were marked with filled black triangle, and the HA sequences of H9N2 vaccines were labelled with red triangle. (B) The NA sequences of prevalent H9N2 isolates were marked with filled black triangles, and the NA sequences of prevalent H9N2 vaccines were labelled with red triangle.

a 313NCS and a 551GSC sites were inserted when compared to a classical Y280 strain, such as A/Duck/HongKong/Y280/97 (Table 2). At the potential glycosylation site 141, there were 29 isolates sharing the same aa sequence VSY with Y280, whereas the aa sequence VTY was present in the rest of the strains (Table 2). The HA genes of the H9 subtype AIVs at positions 109, 161, 163, 191, 198, 202, 203, 146–150, and 232–237 formed an RBS pouch structure (*Matrosovich et al., 2007*). The RBS (aa 232–237) in the HA of the field isolates was NGLMGR, which was highly conservative (Table 2). We observed four mutation types (T/S/L/A) at aa site 198 after aligned with the HA sequence of the recently isolated viruses (Table 2). Notably, mutations were found in four sites that were related to the specific SA of the HA binding, namely S/T at position 143, D/S at 145, T/K at 149, and N/G at 201 (Table 2). These various mutations suggest that the virus has adapted to the host or escaped during vaccination. Leucine (L) was the aa at position 234 (*e.g.*, 226, H3 numbering) in the HA gene sequence of all field strains, indicating its ability to preferentially bind to α-2,6 linked SA, a mammalian species receptor (*Couceiro, Paulson & Baum, 1993*; *Matrosovich et al., 1997*; *Stevens et al., 2006*; *Teng et al., 2016*). Additionally, all isolates carried 225 Glycine (G) in the HA segment (data not shown), which exhibited a human-type receptor-binding property.

## Phylogenetic analysis of NA

The result of the phylogenetic analysis of NA is shown in Fig. 1B. The isolate A/Chicken/Shandong/C82/2017 exhibited high homology with A/Duck/Hong Kong/Y439/1997. Overall, the NA sequence of field isolates somewhat exhibited a genetic distance with that of vaccine strains (Fig. 1B). The H9N2 field virus identities at the nt and deduced aa sequence of NA ranged from 94.4–100% and 94.6–100%, respectively (Fig. S1C). The NA sequence homologies between the isolates and commercial vaccine strain A/Chicken/Guangdong/SS/94, A/Chicken/Shandong/HL/2010, A/Chicken/Jiangsu/WJ57/2012, and A/Chicken/Jiangsu/SZ0207/2013 were 90.7–91.5%, 92.6–93.4%, 92.0–92.7%, and 95.9–97.1%, respectively (Fig. S1D). This result indicated that there was genetic variation between the NA gene of field isolates and the vaccine strains.

The majority of isolates contained seven potential glycosylation sites, which was comparable to that of the Y280-like isolates. The aa sequence of 365GSR/DSR at the potential glycosylation sites was substituted by 399WSG, while an isolate named A/Chicken/Shandong/C54/2017 possessed 10 potential glycosylation sites (plus 2PIQ, 261ISP, and 44PSN) when compared to the classical Y280-like isolates. It was shown that strains A/Chicken/Jiangxi/C19/2017, A/Chicken/Hubei/65/2017, and A/Chicken/Jiangxi /C42/2017 lacked 44PSN aa.

## Antigenicity analysis

To understand the antigenicity of the H9N2 viruses, the supernatant from 45 H9N2-positive samples were selected and inoculated into 9-day-old SPF chicken eggs, and the HA/HI test was performed. The results of the HI test are shown in Fig. 2. The 45 isolates exhibited a positive reaction with the standard serum antibody of H9 AIV, developed by

**Table 2  Amino acid sequence at the receptor binding site and potential glycosylation site.**

| Virus | Amino acid sequence and position of receptor binding site | | | | | | | | | | | | | Potential glycosylation site | | | | | | | | |
|---|---|---|---|---|---|---|---|---|---|---|---|---|---|---|---|---|---|---|---|---|---|---|
| | 143 | 145 | 149 | 153 | 164 | 166-168 | 171 | 181 | 196 | 197 | 198 | 200 | 201 | 29 | 82 | 141 | 218 | 298 | 305 | 313 | 492 | 551 |
| A/Duck/Hong Kong/Y280/97 | S | S | K | D | Q | NNA | I | G | D | T | T | T | N | STE | PSC | VSY | RTF | TTL | VSK | 0 | GTY | 0 |
| A/chicken/Beijing/1/1994 | T | . | . | . | . | D.. | V | . | . | . | V | . | . | ... | ... | .T. | ... | ... | ... | 0 | ... | 0 |
| A/chicken/Hong Kong/G9/1997 | . | . | . | . | . | ... | . | . | . | . | A | . | . | ... | .s. | ... | ... | ... | ... | 0 | ... | GSC |
| A/Chicken/Shanghai/F/98 | . | . | . | . | . | ... | . | . | . | . | A | . | . | ... | ... | ... | ... | ... | ... | CSK | ... | GSC |
| A/duck/Hong Kong/Y439/1997 | T | T | R | N | H | S.S | V | . | . | . | E | . | . | ... | ... | .T. | ... | ... | ... | 0 | ... | GSC |
| A/Quail/Hong Kong/G1/97 | T | T | R | G | . | SGF | . | . | Y | . | E | . | . | ... | ... | .T. | ... | ... | ... | 0 | ... | GSC |
| A/Chicken/Hubei/76/2018 | T | . | . | . | . | ... | . | E | E | . | S | . | . | ... | ... | .T. | 0 | ... | ... | CSK | ... | GSC |
| A/Environment/Jiangxi/E11/2018 | T | . | . | . | . | ... | . | E | E | . | S | . | . | ... | ... | ... | 0 | ... | ... | CSK | ... | GSC |
| A/Chicken/Hubei/22/2018 | . | D | T | N | R | DGN | T | . | . | . | D | R | | ... | ... | .. | 0 | ... | ... | CSK | ... | GSC |
| A/Chicken/Hubei/81/2018 | T | . | . | . | . | ... | . | E | E | . | S | . | . | ... | ... | .T. | 0 | ... | ... | CSK | ... | GSC |
| A/Chicken/Hubei/61/2018 | T | . | . | . | . | ... | . | E | E | . | S | . | . | ... | ... | .T. | 0 | ... | ... | CSK | ... | GSC |
| A/Chicken/Hubei/127/2018 | . | D | T | G | R | .GE | T | . | . | . | D | R | G | ... | ... | .. | 0 | ... | ... | CSK | ... | GSC |
| A/Chicken/Shandong/C84/2018 | . | D | T | G | R | .GD | . | . | . | . | E | R | G | ... | ... | .. | 0 | ... | ... | CSK | ... | GSC |
| A/Duck/Jiangxi/D10/2018 | . | D | T | N | R | DGN | T | . | . | . | D | R | | ... | ... | .. | .. | ... | ... | CSK | ... | GSC |
| A/Chicken/Shandong/C131/2018 | . | D | T | G | R | .GD | . | . | . | . | E | R | G | ... | ... | .. | 0 | ... | ... | CSK | ... | GSC |
| A/Chicken/Hubei/226/2018 | T | . | . | . | . | ... | . | E | E | . | S | . | . | ... | ... | .T. | 0 | ... | ... | CSK | ... | GSC |
| A/Chicken/Hubei/142/2018 | T | . | . | . | . | ... | . | E | E | . | S | . | . | ... | ... | .T. | 0 | ... | ... | CSK | ... | GSC |
| A/Chicken/Hubei/126/2018 | . | D | T | G | R | .GE | T | . | . | . | D | R | G | ... | ... | .. | 0 | ... | ... | CSK | ... | GSC |
| A/Chicken/Hubei/147/2018 | T | . | . | G | . | .G. | . | E | E | . | S | . | . | ... | ... | .T. | 0 | ... | ... | CSK | ... | GSC |
| A/Chicken/Hubei/149/2018 | . | D | T | G | R | .GE | T | . | . | . | D | R | G | ... | ... | .. | 0 | ... | ... | CSK | ... | GSC |
| A/Chicken/Hubei/251/2018 | . | D | T | G | R | .GD | . | . | . | . | E | R | G | ... | ... | .. | 0 | ... | ... | CSK | ... | GSC |
| A/Chicken/Shandong/C54/2018 | . | D | T | G | R | ..D | . | . | . | . | E | R | G | ... | ... | .T. | 0 | ... | ... | CSK | ... | GSC |
| A/Chicken/Hubei/95/2018 | T | . | . | . | . | ... | . | E | E | . | S | . | . | ... | ... | ... | 0 | ... | ... | CSK | ... | GSC |
| A/Chicken/Shandong/C82/2018 | . | D | T | G | R | .GD | . | . | . | . | E | R | G | ... | ... | .T. | 0 | ... | ... | CSK | ... | GSC |
| A/Chicken/Hubei/261/2018 | . | D | T | G | R | .GD | . | . | . | . | E | R | G | ... | ... | .. | 0 | ... | ... | CSK | ... | GSC |
| A/Chicken/Hubei/80/2018 | T | . | . | . | . | ... | . | E | E | . | S | . | . | ... | ... | .T. | 0 | ... | ... | CSK | ... | GSC |
| A/Chicken/Hubei/160/2018 | . | D | T | G | R | .GE | T | . | . | . | D | R | G | ... | ... | ... | 0 | ... | ... | CSK | ... | GSC |
| A/Chicken/Shandong/C107/2018 | . | D | T | G | R | .GD | . | . | . | . | E | R | G | ... | ... | .. | 0 | ... | ... | CSK | ... | GSC |

Peer J

## Table 2 (*continued*)

| Virus | Amino acid sequence and position of receptor binding site | | | | | | | | | | | | | Potential glycosylation site | | | | | | | | |
|---|---|---|---|---|---|---|---|---|---|---|---|---|---|---|---|---|---|---|---|---|---|---|
| | 143 | 145 | 149 | 153 | 164 | 166-168 | 171 | 181 | 196 | 197 | 198 | 200 | 201 | 29 | 82 | 141 | 218 | 298 | 305 | 313 | 492 | 551 |
| A/Chicken/Shandong/C169/2018 | . | D | T | G | R | .GD | . | . | . | E | . | R | G | ... | ... | .. | 0 | ... | ... | CSK | ... | GSC |
| A/Chicken/Jiangxi/C15/2018 | . | D | T | N | R | DGN | T | . | . | D | . | R | . | ... | ... | .. | 0 | ... | ... | CSK | ... | GSC |
| A/Chicken/Shandong/C79/2018 | . | D | T | G | R | .GD | . | . | . | E | . | R | G | ... | ... | .. | 0 | ... | ... | CSK | ... | GSC |
| A/Environment/Jiangxi/E15/2018 | . | D | T | N | R | DGN | T | . | . | D | . | R | . | ... | ... | .. | 0 | ... | ... | CSK | ... | GSC |
| A/Chicken/Jiangxi/C19/2018 | . | D | T | N | R | DGN | T | . | . | D | . | R | . | ... | ... | .. | 0 | ... | ... | CSK | ... | GSC |
| A/Chicken/Shandong/C80/2018 | . | D | T | G | R | .GD | . | . | . | E | . | R | G | .. | ... | .. | 0 | ... | ... | CSK | ... | GSC |
| A/Chicken/Jiangxi/C16/2018 | . | D | T | N | . | DGN | T | . | . | D | . | R | . | ... | ... | .. | 0 | ... | ... | CSK | ... | GSC |
| A/Chicken/Hubei/146/2018 | T | . | . | . | . | ... | . | E | E | . | S | . | . | ... | ... | .T. | 0 | ... | ... | CSK | ... | GSC |
| A/Chicken/Hubei/78/2018 | T | . | . | . | . | ... | . | E | E | . | S | . | . | ... | ... | .T. | 0 | ... | ... | CSK | ... | GSC |
| A/Chicken/Jiangxi/C11/2018 | . | . | . | . | . | ... | T | . | . | . | A | . | . | ... | ... | .. | 0 | ... | ... | CSK | ... | GSC |
| A/Duck/Jiangxi/D13/2018 | . | . | . | . | . | ... | . | E | E | . | A | . | . | ... | ... | .. | 0 | ... | ... | CSK | ... | GSC |
| A/Chicken/Hubei/131/2018 | T | . | . | . | . | ... | . | E | E | . | S | . | . | ... | ... | .T. | 0 | ... | ... | CSK | ... | GSC |
| A/Chicken/Hubei/92/2018 | . | . | . | . | . | ... | T | . | . | . | A | . | . | ... | ... | .. | 0 | ... | ... | CSK | ... | GSC |
| A/Chicken/Jiangxi/C14/2018 | T | . | . | . | . | ... | . | E | E | . | S | . | . | ... | ... | .T. | 0 | ... | ... | CSK | ... | GSC |
| A/Chicken/Hubei/63/2018 | . | . | . | . | . | ... | . | E | E | . | A | . | . | ... | ... | .. | 0 | ... | ... | CSK | ... | GSC |
| A/Chicken/Shandong/C126/2018 | . | D | T | G | R | .GD | . | . | . | E | . | R | G | ... | ... | .T. | 0 | ... | ... | CSK | ... | GSC |
| A/Chicken/Hubei/65/2018 | T | . | . | . | . | ... | . | E | E | . | S | . | . | ... | ... | .T. | 0 | ... | ... | CSK | ... | GSC |
| A/Chicken/Hubei/107/2018 | T | . | . | . | . | ... | . | E | E | . | S | . | . | ... | ... | .T. | 0 | ... | ... | CSK | ... | GSC |
| A/Duck/Jiangxi/D1/2018 | T | . | . | . | . | ... | . | E | E | . | S | . | . | ... | ... | .T. | 0 | ... | ... | CSK | ... | GSC |
| A/Chicken/Hubei/90/2018 | . | D | T | N | R | DGN | T | . | . | D | . | R | . | ... | ... | .. | 0 | ... | ... | CSK | ... | GSC |
| A/Chicken/Hubei/169/2018 | . | D | T | G | R | .GD | . | . | . | E | . | R | G | ... | ... | .. | 0 | ... | ... | CSK | ... | GSC |
| A/Chicken/Hubei/77/2018 | T | . | . | . | . | ... | . | E | E | . | S | . | . | ... | ... | .T. | 0 | ... | ... | CSK | ... | GSC |
| A/Chicken/Jiangxi/C42/2018 | . | D | T | G | R | .GD | . | . | . | E | . | R | G | ... | ... | .. | 0 | ... | ... | CSK | ... | GSC |
| A/Goose/Jiangxi/G1/2018 | T | . | . | . | . | ... | . | E | E | . | S | . | . | ... | ... | .T. | 0 | ... | ... | CSK | ... | GSC |
| A/Chicken/Hubei/152/2018 | . | D | T | N | R | DGN | T | . | . | D | . | R | . | ... | ... | .. | 0 | ... | ... | CSK | ... | GSC |
| A/Chicken/Hubei/262/2018 | . | . | . | . | . | ... | T | . | . | . | A | . | . | ... | ... | .. | 0 | ... | ... | CSK | ... | GSC |

**Notes.**

"." represents that the site at which the amino acid is identical to A/Duck/Hong Kong/Y280/97 (H9N2) virus; "0" indicates a site at which the potential glycosylation site was absent.

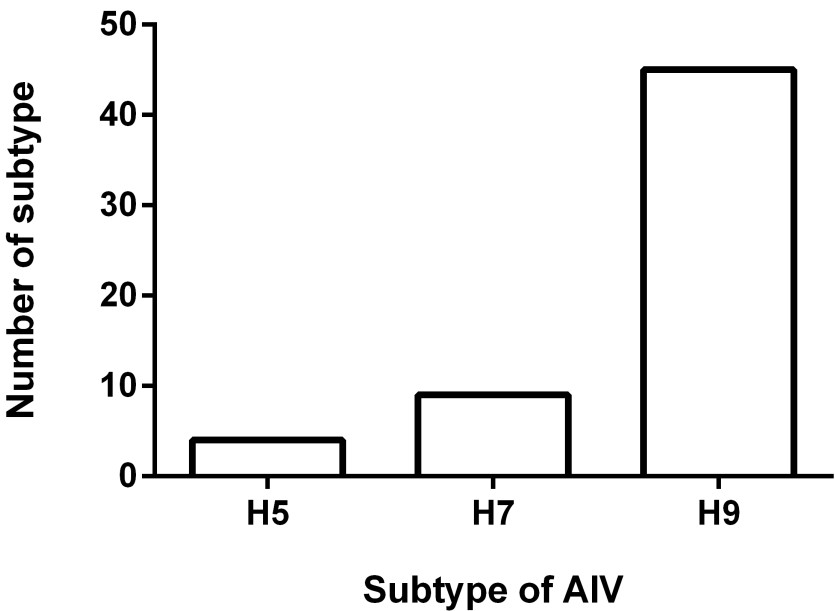

**Figure 2** **Result of HI test.** Fifty five H9N2 AIV isolates were used as antigen for the HI test using standard antiserum against H5, H7 and H9 subtype AIV, respectively. The number of samples in which the virus reacted with H5, H7 and H9 subtype antiserum was shown.

Harbin Veterinary Research Institute (HVRI), Chinese Academy of Agricultural Sciences (CAAS), China. The HI titer ranged from 7log2 to 10log2, and the averaged titer was 8.2log2. Notably, four viruses reacted with the positive serum of the H5 subtype AIV, as demonstrated by an averaged HI titer of 6Log2. We found that nine field viruses reacted with the H7 AIV positive serum, with a titer range of 4log2 to 5log2. Interestingly, all viruses with the ability to react with the H5 positive serum could also react with the H7 positive serum. Four viruses reacted well with the H5, H7, and H9 antibodies, and five virus strains could only react with the H7 and H9 antibodies.

## DISCUSSION

H9N2 viruses in the Y280 lineage have been spreading across China in recent years (*Xue et al., 2014*; *Cong et al., 2017*; *Wang et al., 2011*). Previous reports indicated that the major genotypes of H9N2 AIVs in China could be divided into five series: BJ/94-, G1-, BG-, F/98-, and Aq-series (*Sun et al., 2010*). In this study, a total of 67 out of 77 AIVs were detected to be H9N2 subtypes. A Y280 strain was selected as the reference strain to analyze the evolution of the HA and NA segments of H9N2 field isolates. Using the HA nucleotide sequence, a biological phylogenetic tree was constructed, and the phylogenetic analysis showed that the H9N2 isolates shared a relatively closer genetic relationship with the Y280 strain, and H9N2 isolates and the strain SH98 in the NA fragment showed the closest relatedness. All H9N2-positive samples were collected from Hubei, Jiangxi, and Shandong, but no AIV was detected in samples from Zhejiang. We hypothesized that the better biosafety management of farms and more effective animal disease control measures implemented in Zhejiang

may partially explain the absence of AIV in the samples we detected. As shown in the phylogenetic tree in Fig. 1, the HA nucleotide sequence of H9N2 viruses from Shandong province differed significantly from the prevalent reference strains. The geographic location of Shandong province may explain this difference, since Shandong is located in an area along the coastline of the Bohai Sea where bird migration is more frequent. This likely increases the chance of reassessment of domestic H9N2 viruses with other subtype AIVs, leading to the genetic variation of the H9N2 viruses. Poyang Lake is located in Jiangxi province, meaning that a large number of waterfowls, *e.g.,* ducks and geese, breed in this area (*Li et al., 2017*). It has been well-documented that H9N2 AIV provides internal genes to other subtype AIVs such as H7N9 virus, resulting in a new generation of reassortant viruses that pose an increased threat to public health (*Pu et al., 2015*; *Gong et al., 2014*; *Pu et al., 2021*; *Su et al., 2015*).

Inactivated vaccines against H9N2 avian influenza have been used in China for decades. However, active surveillance has shown that the H9N2 virus was frequently detected in samples collected from poultry farms, LBMs, and the environment, indicating that the currently-used vaccines for H9N2 avian influenza are not as effective as needed, and therefore need to be updated. The findings of this study supported this observation. These results suggest that new candidate vaccines that correspond better with dominantly circulating H9N2 AIVs need to be developed (*Liu et al., 2002*).

In our survey, all H9N2 isolates had cleavage sites of PSRSSRG/L, which was a characteristic of LPAIV. Potential glycosylation sites at the HA of H9N2 AIV were at aa positions 8, 29, 82, 141, 218, and 298 (H9 numbering). Our result showed that, compared to the Y280 strain, the aa at the glycosylation site of 313 CSK and 551 GSC was inserted, while the aa of 218 RTF was deleted. The glycosylation site at 402, which was reportedly related to the characteristics of H9N2 viruses (*Kandeil et al., 2017*), was not found in any virus analyzed in this study. N-linked glycosylation sites were confirmed to play critical roles in viral infectivity and host cell immune response. Particularly, the glycosylation site in the HA cleavage site affected the cleavage capability of HA by protease, leading to a change in virulence (*Owen et al., 2007*). The RBS is of great significance for host cellular receptor specificity and plays an important role in the generation of reassortant influenza viruses (*Wang, 2016*).  The HA RBS bears a certain degree of variation, mainly in 198 V/T/S/L/A and 168 A/S/N/E/D/F. RBS is closely related to the host cell and virulence, and we speculate that it may have an intimate relationship with the escape immune defense mechanism that AIV induces in hosts after vaccination (*Liu et al., 2002*). HA is the most abundant protein on the surface of the virion, and exhibits specific binding affinities for the different SA-linked glycoproteins that are expressed on the cell–surface receptors. AIVs preferentially bind to SA linked to the terminal oligosaccharide by an α-2,3-linked bond (generally referred to as the avian receptor), while human influenza virus strains favor the α-2,6-linked SA receptor (generally referred to as the human receptor) (*Matrosovich et al., 1997*; *Stevens et al., 2006*). It was previously found that the intestinal epithelial cells of poultry were mainly distributed with α-2,3 receptors (*Ito et al., 2000*), while the epithelial cells of human upper respiratory tracts mainly had α-2,6 receptors (*Couceiro, Paulson & Baum, 1993*). Therefore, the prerequisite for AIV infection in humans is a mutated receptor

binding site of the HA protein, with the ability to bind to the α-2,6-linked SA receptor, either naturally or artificially. A switch from a-2,3 to a-2,6 binding specificity requires several HA mutations, and the mechanism of this situation is complex (*Shi et al., 2014*). There are α-2,3 receptors on the human epithelial cells of the lower respiratory tract, alveoli, and lung macrophages (*Shinya et al., 2006*; *Van Riel et al., 2006*). Notably, the leucine (Leu) residue at position 226 of the HA protein creates a more hydrophobic environment than that of the HA of AIV, which likely accounts for the absence of water molecules in this region. Furthermore, the shortest distance between the side-chains of Leu-226 and C6 of the ligand are too long to generate a significant hydrophobic interaction to facilitate the binding of the α-2,6-linked ligand. In contrast, in unliganded avian HA, Gln-226 coordinates a water molecule that is very close to the position occupied by Gal-2 in the human HA-human receptor complex. This suggests that human HAs benefit from acquiring Leu at position 226 because human receptors can bind without the need to displace water. The most obvious explanation for the weaker binding of avian receptors, also observed in binding assays (*Matrosovich et al., 2000*), lies in the positioning of the hydrophobic Leu-226 underneath where glycosidic oxygen would normally occupy the α-2,3-linked avian receptor. Avian α-2,3-linked receptors tend to bind in a trans conformation at the Sia-1-Gal-2 glycosidic bond, while human α-2,6-linked receptors adopt a cis conformation, meaning that one of the loan pairs of electrons of the glycosidic oxygen in the avian receptor is oriented toward Gln-226, rather than away from the protein that is toward the solvent, as in the case of the human receptor (*Eisen et al., 1997*; *Liu et al., 2009*). Additionally, because HA Q234L (H9 numbering) holds a high affinity with α-2,6 SA of human mucous cells, they have the ability to infect humans.

The location and number of HA glycosylation sites and the RBS on influenza viruses were determined via virus genomic sequence. The replication of the virus was initiated and promoted by RNA-dependent RNA polymerase. The lack of self-correction of polymerase resulted in an increased rate of genetic mutations in AIV when compared to other viruses (*Guan et al., 1999*; Liu et al., 2003; *Zhang et al., 2016*). If mutations occurred on HA glycosylation sites and/or RBS, new amino acids were often introduced or deleted. If the newly-occurring oligosaccharide chain was near the glycosylation site due to gene mutations, the original structure of the antigenic determinant was destroyed by the appearance of the oligosaccharide chain, and the antibody produced by the original vaccine was not neutralized, triggering new epidemics (*Zhang et al., 2018*).

The results in this study showed that the NA gene sequence of H9N2 field isolates had the highest homology with the classical strain SH98, while the HA gene had the highest homology with the Y280 strain. This result indicated that the field isolates were reassortant viruses. The NA gene of the field isolates exhibited more significant genetic variations than the prevalent and vaccine strains used in recent years, and formed a distinct evolutionary branch. The circulating isolates examined in our survey were compared to the classical strains in the aa 63-65 in the stem of NA, and showed different potential glycosylation sites of the field isolates. Most field isolates had seven potential glycosylation sites, but the classical strain had lost 61NTE and had a new 365GSR site introduced. Additionally, our data showed three strains that lacked 44PSN, and one isolate, A/Chicken/Shandong/C54/2017,

that had nine glycosylation sites and a new glycosylation site 2PIQ, when compared to the classical H9N2 AIV isolates. The function of NA was tantamount when cutting off the connection between HA and host cells and when facilitating the release of progeny virus. The loss of the NA stem led to altered viral virulence and enhanced infectivity to host cells (*Qi & Lu, 2006*). Our findings underscored the necessity to continuously update H9N2 AIV vaccines with the NA from prevalent strains.

The results of the HI assay depicted in this study suggested that the HA and NA of H9N2 field isolates were constantly evolving and reassorting. Our HI test results suggest the possibility of reassortment between the H9 and H5 HA gene fragments. The H7 subtype AIV could not be ruled out, which may have caused the cross immune response detected in this study. Given that the H9N2 virus could provide internal genes for H7 and/or H5 subtype HPAIVs, resulting in a reassortant virus that may have the ability to infect humans, the mutation of the H9N2 virus should not be ignored. In addition, nationwide compulsory vaccination programs that use trivalent vaccines, namely (H5) Re-11/Re-12 + (H7) Re-3 inactivated vaccines, are currently implemented in China for the prevention and control of H5 and H7 subtype HPAIV. The compulsory vaccination campaign played an important role in HPAIV prevention and control in China. Nevertheless, the H5 and H7 subtype HPAIVs are continuously evolving under the pressure of immunization. Therefore, continuous monitoring of H9N2 AIVs and the timely updating of vaccines are of great importance for H9N2 subtype influenza prevention and control. Our findings confirmed that LBMs are an ideal place for H9N2 and other subtype AIV reassortment, and LBM management should also be improved.

# ACKNOWLEDGEMENTS

We are grateful to the anonymous reviewers for providing insightful comments on this paper.

## Funding
This work was supported by the National Key Research and Development Program: 2016YFD0501609. The funders had no role in study design, data collection and analysis, decision to publish, or preparation of the manuscript.

## Grant Disclosures
The following grant information was disclosed by the authors:
The National Key Research and Development Program: 2016YFD0501609.

## Competing Interests
The authors declare there are no competing interests.

## Author Contributions

- Xiaoyi Gao, Naidi Wang, Yuhong Chen and Yuanhui Huang, Yang Liu, Fei Jiang, Jie Bai and Lu Qi performed the experiments, analyzed the data, prepared figures and/or tables, authored or reviewed drafts of the paper, and approved the final draft.
- Xiaoxue Gu performed the experiments, analyzed the data, prepared figures and/or tables, and approved the final draft.
- Shengpeng Xin, Yuxiang Shi and Chuanbin Wang conceived and designed the experiments, authored or reviewed drafts of the paper, and approved the final draft.
- Yuliang Liu conceived and designed the experiments, performed the experiments, analyzed the data, prepared figures and/or tables, authored or reviewed drafts of the paper, and approved the final draft.

## Field Study Permissions

The following information was supplied relating to field study approvals (i.e., approving body and any reference numbers):

Field sample collection was approved by China Animal Disease Control Center [CADC-Surveillance (2018) No. 53].

## DNA Deposition

The following information was supplied regarding the deposition of DNA sequences:

The HA and NA sequences of H9N2 subtype avian influenza virus are available at GenBank: HA gene; MW467319 to MW467367 and NA gene; MW466644 to MW466692.

## Data Availability

The data are available in the Supplementary File.

## Supplemental Information

Supplemental information for this article can be found online at http://dx.doi.org/10.7717/peerj.12512#supplemental-information.

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
