# Peer review of "Sequence characteristics and phylogenetic analysis of H9N2 subtype avian influenza A viruses detected from poultry and the environment in China, 2018"

_PeerJ, doi:10.7717/peerj.12512_

## Round 0.1 · original submission · Major Revisions

Experts have reviewed your submission and raised several concerns that preclude it from acceptance. Please address their comments line by line in your rebuttal.

Reviewer 1 ·

Basic reporting

This manuscript summarizes AI surveillance efforts in different provinces in China. While the results provided are interesting and conclusions can be drawn that can benefit the prevention strategies currently used, the manuscript needs some work to be of publishable quality. In the materials and methods section, additional methodologies need to be explained. The results need to be discussed in the discussion section rather than in the results section. Some clarification of the surveillance design is also needed. The organization of the figures need to be revised. I suggest adding to the manuscript a clear hypothesis and objectives to give organization to the report.

Experimental design

Clearly stated hypothesis and objectives would improve the organization of this manuscript. A revision of the materials and methods and clarification of the process is needed e.g. were the detections made from isolates or RT-PCR from swabs? This is not clear in the text.

Validity of the findings

This manuscript summarizes AI surveillance efforts in different provinces in China. While the results provided are interesting and conclusions can be drawn that can benefit the prevention strategies currently used, the manuscript needs some work to be of publishable quality

Additional comments

PeerJ #50819
Title: Sequence characteristics and phylogenetic analysis of H9N2 subtype avian influenza A viruses detected from poultry and environment in China, 2018
Authors: Xiaoyi Gao, Naidi Wang, Yuhong Chen, Yuanhui Huang, Xiaoxue Gu, Yang Liu, Fei Jiang, Jie Bai, Lu Qi, Shengpeng Xi , Chuanbin Wang, Yuxiang Shi, Yuliang Liu

This manuscript summarizes AI surveillance efforts in different provinces in China. While the results provided are interesting and conclusions can be drawn that can benefit the prevention strategies currently used, the manuscript needs some work to be of publishable quality. In the materials and methods section, additional methodologies need to be explained. The results need to be discussed in the discussion section rather than in the results section. Some clarification of the surveillance design is also needed. The organization of the figures need to be revised. I suggest adding to the manuscript a clear hypothesis and objectives to give organization to the report.

Abstract

-I would be more specific in the regions where this surveillance strategy occurred.
-Was this active surveillance
-Line 44. Says “salivary acid” and should say “sialic acid”
-Line 45. It would be interesting to say why those genetic variations are important

Introduction

Line 60. This is based on their virulence “in chickens”
Line 63. As stated, you are giving ostriches as an example of wild birds, please correct.
Line 67. Poultry is plural, please change poultries to poultry. Poultry can also be used for a single bird.
Lines 97 to 99. Please rephrase, not understandable on its current form
The introduction is very wide. This introduction should introduce what the manuscript will be covering:
-An active surveillance strategy using HA gene for H9N2 viruses
-and justify why that is done in which region and using which methods.

Materials and methods
Line 111. PBS + AB is not viral transport media, please erase “viral transport media”
Line 113. Not sure which kind of examination needs to be done to the samples but this sentence doesn’t make sense
Table 1. Please eliminate the province column, since the sample nomenclature states the province from where the samples were collected
Line 117 to 135. The fragment size is not reported for HA and NA
line 142. Please eliminate ‘the” to “The GenBank”
Line 144 to 146. Since all AIV’s are hemagglutinating, how can you evaluate antigenicity using HA property? If you use different antibodies to different HA types you could evaluate antigenicity, is that what was done? Please be clearer and add additional data in the way in which the viruses were evaluated on their antigenicity.

Results

Line 149-151. There is a confusion between the tests done during the surveillance. Are the swabs used for RT-PCR, virus isolation, all? Please be clear on the manuscript.
Figure 1 is referenced in the text after figure 2, you need to change the numbering.
Figure 2. The legends seem materials and methods, the legend should explain what the graph is showing and stand alone in terms of explaining what is shown, is there any stats calculation?
The supplementary figure 1 is not included in this document.
Line 157, 161, 199, etc. Were these viruses isolated or detected by molecular biology. If the latter is correct don’t use the word isolation. If they were isolated add the isolation procedure on the materials and methods section and make sure to be clear in reporting.
Line 163. Were nt and aa explained as acronyms before?
Line 169. Supplementary figures were not included in this version of the manuscript
Line 184 to 189. These elements are part of the discussion not result section.
Lines 191 to 193. This seems to be part of the discussion, please change “salivary” to “sialic”
Line 212 to 214. This should be part of the discussion section

Discussion

Line 235 to 236. Please rephrase.
Please clarify the isolation statements.

Annotated reviews are not available for download in order to protect the identity of reviewers who chose to remain anonymous.

Reviewer 2 ·

Basic reporting

The continous sampling,testing of birds and poultry and the sequencing of positive samples are important to prevent and control AIV and to detect new reassortants as early as possible. The continous evolution of AIV should be monitored and the obtained data should be made public.
The manuscript is well written and easy to follow. The outline of the study is well explained. The conclusion drawn from the results are comprehensible.

Experimental design

line 109: "A total of 1529 oropharyngeal and cloacal swab samples were collected."
line 159: "one from environment"
How many environmental samples were tested? No information is found in the material & method part.
line 216-218: Could the authors please state the HI titer values obtained against H9(N2)?
line 219-220: Could the authors please state the HI titer values obtained against H7(N9)?

Validity of the findings

line 291-292: Please discuss whether previous AIV-infections could account for immunity against H5 and H7 Antigens.

Additional comments

The presented mansucript needs some formatting and re-writing.
line 72-73: The authors could also mention H5N6 (Cui et al., 2020; doi.org/10.1080/22221751.2020.1797542)
line 151-152: "83 isolates were detected positive for AIV, and 87.02% (67/77)" - Please state why 6 Isolates were not subtyped? Subytping was not successful?
line 153/216: Fig.2 - Following the text in line 216 - These figure should show the results of the HI assay and not distribution of H7N9, H3N6, H4N6 positive samples. The figure labelling also says:"The samples were collected and detected by RT-PCR. RT-PCR positive samples were
submitted for sequencing, and the sequences were analyzed by BLAST to determine the
subtypes of AIV. Total number of each subtype was shown." - Please rewrite the labelling and the citation in the manuscript.
line 237: "phylogenetic tree in Fig. 2" - Figure 2 does not show a phylogenetic tree
line 264-265: "HA Q234L (H9 numbering) and HA Glutamine (Q) 234
265 lysine (L) (H9 numbering)" - double mentioning of the exact same Mutation
line 296: ideal instead of idea
Tab. 1: the sample number 4 has been used twice whereas number 2 is missing.

---

## Round 0.2 · Minor Revisions

While I and the reviewer agree the manuscript clarity has improved, there are still some pending revisions that are needed before acceptance. In addition, the evidence for the higher likelihood of human transmission (via sialic acid) is based on a single reference, there is more explanation here needed to even convince me as the editor that this truly translates to greater risk for humans. Hence, either a more deeper discussion is needed with more than just that reference (more references and explanation of that mechanism) or further evidence is needed.

Finally, there are still some pending English language revisions that are needed, I strongly suggest the authors do that or seek help with that as it will help reduce the time in review.

Best wishes,
Sharif

Reviewer 1 ·

Basic reporting

This is the second round of reviews to this paper. The manuscript was considerably improved in quality. However, there re still some issues in phrase construction and english. i would advise authors to look for a native English writer to review and edit the manuscript for clarity in order to take it to publishable quality. I did not see a clear hypothesis and objectives added to this manuscript

Comments:
Line 66. Ostriches are considered domestic poultry unless you refer to wild ostriches. I would either erase that or refer to a larger group.
Line 98. Were inoculated
Line 151. Please revise this sentence “Embryos that died after…”
Line 154. That centrifugation won’t clarify allantoic fluid effectively from blood, that affects the viral titer

Experimental design

Line 154. That centrifugation speed and time won’t clarify allantoic fluid effectively from red blood cells, affecting the viral titer

Validity of the findings

No comments

Additional comments

This is the second round of reviews to this paper. The manuscript was considerably improved in quality. However, there re still some issues in phrase construction and english. i would advise authors to look for a native English writer to review and edit the manuscript for clarity in order to take it to publishable quality. I did not see a clear hypothesis and objectives added to this manuscript

---

## Round 0.3 · accepted · Accept

Thank you for addressing the reviewer comments and providing further references documenting the earlier identified statements. In agreement with the reviewer, I find your manuscript acceptable for publication. Best wishes in your future research.
Sharif

Reviewer 1 ·

Basic reporting

Manuscript was greatly improved

Experimental design

ok

Validity of the findings

ok

Additional comments

No additional comments